# Extensive alternative splicing transitions during postnatal skeletal muscle development are required for calcium handling functions

Amy E Brinegar[1,2], Zheng Xia[1,3†], James Anthony Loehr[4], Wei Li[1,3], George Gerald Rodney[4], Thomas A Cooper[1,2,4]*

[1]Department of Molecular and Cellular Biology, Baylor College of Medicine, Houston, United States; [2]Department of Pathology and Immunology, Baylor College of Medicine, Houston, United States; [3]Division of Biostatistics, Dan L Duncan Cancer Center, Baylor College of Medicine, Houston, United States; [4]Department of Molecular Physiology and Biophysics, Baylor College of Medicine, Houston, United States

**Abstract** Postnatal development of skeletal muscle is a highly dynamic period of tissue remodeling. Here, we used RNA-seq to identify transcriptome changes from late embryonic to adult mouse muscle and demonstrate that alternative splicing developmental transitions impact muscle physiology. The first 2 weeks after birth are particularly dynamic for differential gene expression and alternative splicing transitions, and calcium-handling functions are significantly enriched among genes that undergo alternative splicing. We focused on the postnatal splicing transitions of the three calcineurin A genes, calcium-dependent phosphatases that regulate multiple aspects of muscle biology. Redirected splicing of calcineurin A to the fetal isoforms in adult muscle and in differentiated C2C12 slows the timing of muscle relaxation, promotes nuclear localization of calcineurin target Nfatc3, and/or affects expression of Nfatc transcription targets. The results demonstrate a previously unknown specificity of calcineurin isoforms as well as the broader impact of alternative splicing during muscle postnatal development.
DOI: https://doi.org/10.7554/eLife.27192.001

*For correspondence:
tcooper@bcm.edu

Present address: †Department of Molecular Microbiology and Immunology, Oregon Health and Science University, Portland, United States

Competing interests: The authors declare that no competing interests exist.

## Introduction

There is a 50-fold increase in body weight during murine postnatal development, 50% of which is contributed by skeletal muscle (*Allen et al., 1979*). Skeletal muscle tissue undergoes dynamic remodeling after birth to transition to the functional requirements of adult tissue. While embryonic development of skeletal muscle and regeneration in adult skeletal muscle has been extensively studied, the physiological transitions of postnatal muscle are poorly understood (*Aulehla and Pourquié, 2008*; *Braun and Gautel, 2011*; *Buckingham et al., 2003*; *Dubrulle and Pourquié, 2004*). In rodents, as in humans, skeletal muscle at birth is immature with low functionality as illustrated by poor mobility of newborns. After a period of active proliferation of myogenic progenitor satellite cells and fusion to form myofibers during late embryonic development, myofiber number remains constant after birth. Postnatal skeletal muscle growth is primarily by myofiber hypertrophy and fusion of proliferating satellite cells that is limited to within the first few weeks after birth (*Ontell et al., 1984*; *White et al., 2010*). Satellite cells make up approximately 11% of muscle nuclei at postnatal day 14, but by week 17, the fraction of satellite cell nuclei drops to 3% (*Ontell et al., 1984*) reflecting their transition from contributing to myofiber growth to quiescent adult muscle stem cells

(*Relaix and Zammit, 2012*). The tibialis anterior muscle cross-sectional area increases seven-fold from postnatal day 1 (PN1) to PN28 in mice with a concomitant five-fold increase in maximal isotonic force (*Gokhin et al., 2008*). Importantly, while the increased isotonic force is due primarily to increased muscle size, there is a six-fold increase in intrinsic mechanical function from PN1 to PN28 that is size-independent. The basis for the increased intrinsic function is not completely understood but correlates best with increased myofibril size, a transition of myosin heavy chain isoforms, and changes in metabolism and calcium handling (*Gokhin et al., 2008*). T-tubules and sarcoplasmic reticulum, the cellular structures required for excitation contraction coupling, rapidly mature within the first 3 weeks after birth (*Franzini-Armstrong, 1991*). The majority of skeletal muscles in mice switch from slow, type I fibers to fast, type II fibers (*Agbulut et al., 2003*). In addition to the differences in metabolism, fast and slow fibers have different cytosolic calcium concentrations of 30 nM and 50–60 nM, respectively (*Gailly et al., 1993*).

While changes in gene expression are well-established mediators of skeletal muscle hypertrophy and atrophy the role of different protein isoforms generated by alternative splicing in muscle physiology has not been extensively explored. We hypothesize that there is a substantial role for protein isoform transitions produced by coordinated alternative splicing in the transition to adult skeletal muscle physiology. Alternative splicing generates proteome diversity including isoforms with tissue specific or developmental stage-specific functions (*Giudice et al., 2014*; *Kalsotra et al., 2008*; *Merkin et al., 2012*). Ninety-five percent of human intron-containing genes are alternatively spliced; however, opinions differ with regard to the extent to which this huge diversity is regulated to provide a functional outcome (*Tress et al., 2017*; *Pickrell et al., 2010*; *Gonzàlez-Porta et al., 2013*). Several studies have shown that the majority of alternative splicing is not conserved; however, alternative exons that are tissue-specific or regulated during periods of physiological change show high levels of conservation of the variable protein segment, suggesting functional importance; functionality is further supported when the timing of the splicing transition is also conserved (*Kalsotra et al., 2008*; *Barbosa-Morais et al., 2012*; *Ellis et al., 2012*). For example, alternative splicing transitions during postnatal development of both brain and heart have been associated with functional consequences (*Giudice et al., 2014*; *Dillman et al., 2013*).

While a number of alternative splicing transitions during skeletal muscle postnatal development have been identified, little is known regarding the functional consequences (*Azim et al., 2012*; *Buck et al., 2010*; *Grande et al., 2003*; *Lu et al., 2008*; *Ohsawa et al., 2011*). Recent reports demonstrate functional consequences using knock out of alternative exons to force expression of specific isoforms of Ca$_V$1.1 or titin resulting in reduced contractile force and substantial histopathology, respectively (*Sultana et al., 2016*; *Charton et al., 2016*). Simultaneous reversion to fetal splicing patterns of four vesicular trafficking genes in adult mouse skeletal muscle demonstrated the requirement for the adult isoforms in myofiber structure and physiology (*Giudice et al., 2016a*). Reversion to fetal splicing patterns in skeletal muscle is a hallmark of myotonic dystrophy and results in altered function including myotonia due to failure to express the adult isoform of the *CLCN1* mRNA (*Mankodi et al., 2002*; *Charlet-B et al., 2002*). The full extent to which alternative splicing contributes to normal postnatal muscle development remains unknown since its role has not been systematically examined on a genome-wide scale.

Among a large number of RNA-binding proteins that regulate alternative splicing, the Muscleblind-like (MBNL) and CUG-BP Elav-like family (CELF) families are the best characterized for regulating splicing transitions during postnatal development in heart and skeletal muscle (*Giudice et al., 2014*; *Kalsotra et al., 2008*; *Lin et al., 2006*). Disruption of MBNL and CELF RNA processing activities by the repeat containing RNAs expressed from microsatellite expansions cause the pathogenic effects in myotonic dystrophy (*Lee and Cooper, 2009*). MBNL and CELF families regulate separate as well as overlapping subsets of alterative splicing events and most often show antagonistic regulation of the shared splicing events (*Kino et al., 2009*; *Wang et al., 2015*).

We performed a systematic analysis of genome-wide gene expression and alternative splicing transitions in mouse gastrocnemius muscle by RNA-seq of five time points between embryonic day 18.5 (E18.5) and adult. The results show extensive regulation of both gene expression and alternative splicing that is particularly active within the first 2 weeks after birth. Of the transitions that occur between E18.5 and adult, 55% and 56% of genes that undergo alternative splicing or differential expression, respectively, occur between postnatal day 2 (PN2) and PN14. Interestingly, 58% of the splicing transitions that occur between PN2 and PN14 show little change before and after these

time points identifying a subset of splicing transitions that are not contiguous with ongoing fetal transitions, but rather are limited to the first 2 weeks after birth. The genes that undergo differential gene expression and alternative splicing show minimal overlap suggesting independent mechanisms of transcriptional and post-transcriptional regulation. Differentially expressed genes were enriched for mitochondrial functions while genes that undergo alternative splicing transitions were enriched for calcium handling, cell-cell junction, and endocytosis. We show that more than 50% of the splicing transitions tested were conserved between mouse and human with regard to the direction and timing of the transition strongly suggesting functional significance. We used morpholino oligonucleotides to re-direct splicing of all three calcineurin A genes (*Ppp3ca*, *Ppp3cb*, and/or *Ppp3cc*) to the fetal isoforms in adult mouse flexor digitorum brevis (FDB) muscle and differentiated C2C12 myotubes. Ex vivo analysis of FDB muscle demonstrated an effect on contractile properties and calcium handling. Redirected splicing to the fetal isoforms of calcineurin A in differentiated C2C12 myotubes caused nuclear localization of Nfatc3. These results identify protein isoform transitions that occur during postnatal skeletal muscle development and demonstrate previously unknown isoform-specific functional requirements for activation of calcineurin A transcriptional targets.

## Results

### Transcriptome changes predominate within the first 2 weeks of postnatal skeletal muscle development

To identify changes in gene expression and alternative splicing during postnatal skeletal muscle development, we performed RNA-seq using RNA from gastrocnemius muscle at E18.5, PN2, PN14, 28, and adult (22 weeks) including a biological replicate for PN14. Males were used for all time points except E18.5 for which both male and female animals were used. We obtained >160 million 100 bp paired end reads per sample with at least 92% of reads mapping to the mouse genome (*Table 1*). The PN14 biological replicates revealed strong correlations for both gene expression and alternative splicing indicating high levels of reproducibility ($r^2 = 0.99$ and 0.90, respectively, *Figure 1A and B*). Splicing transitions predicted by RNA-seq were validated by RT-PCR by comparing the change in percent spliced in (ΔPSI) identified by RNA-seq and by RT-PCR between PN2 and PN28 ($r^2 = 0.80$, *Figure 1C–E*). The overall results indicate that our RNA-seq data reflects the transcriptome changes occurring in vivo during postnatal skeletal muscle development.

We identified 4417 genes showing differential expression (±2.0 fold change) and 721 events showing differential splicing (ΔPSI of ±15%) comparing E18.5 to adult skeletal muscle (samples from *Table 1*). For both gene expression and alternative splicing, the interval between PN2 and PN14 was the most dynamic time period with regard to the numbers of genes undergoing transitions. The analysis is affected by the differences in interval length between times points (E18.5 to PN2 vs. PN28 to adult) but even after correcting for differences in interval duration, the largest number of genes change expression between PN2 and PN14 (*Figure 2—figure supplement 1A*). From E18.5 to adult time points, 56% of differential gene expression changes occurred between PN2 and PN14 (3315 genes) while expression of 636 genes (11%) changed between PN14 and PN28, 481 genes (8%)

**Table 1.** Mouse tissue samples used for RNA-seq.
Samples were pooled for E18.5 and PN2 to obtain sufficient quantities of RNA.

| Sample | Tissue | Age | Sex | # of mice | Mapped reads | % mapped |
|---|---|---|---|---|---|---|
| E18.5 | Gastrocnemius | E18.5 | M/F | 21 | 377,538,308 | 93.00% |
| PN2 | Gastrocnemius | PN2 | M | 6 | 381,045,805 | 92.20% |
| PN14 Replicate #1 | Gastrocnemius | PN14 | M | 1 | 321,001,586 | 92.60% |
| PN14 Replicate #2 | Gastrocnemius | PN14 | M | 1 | 379,096,597 | 93.20% |
| PN28 | Gastrocnemius | PN28 | M | 1 | 377,248,209 | 93.30% |
| Adult | Gastrocnemius | 22 weeks | M | 1 | 381,729,569 | 93.50% |

DOI: https://doi.org/10.7554/eLife.27192.004

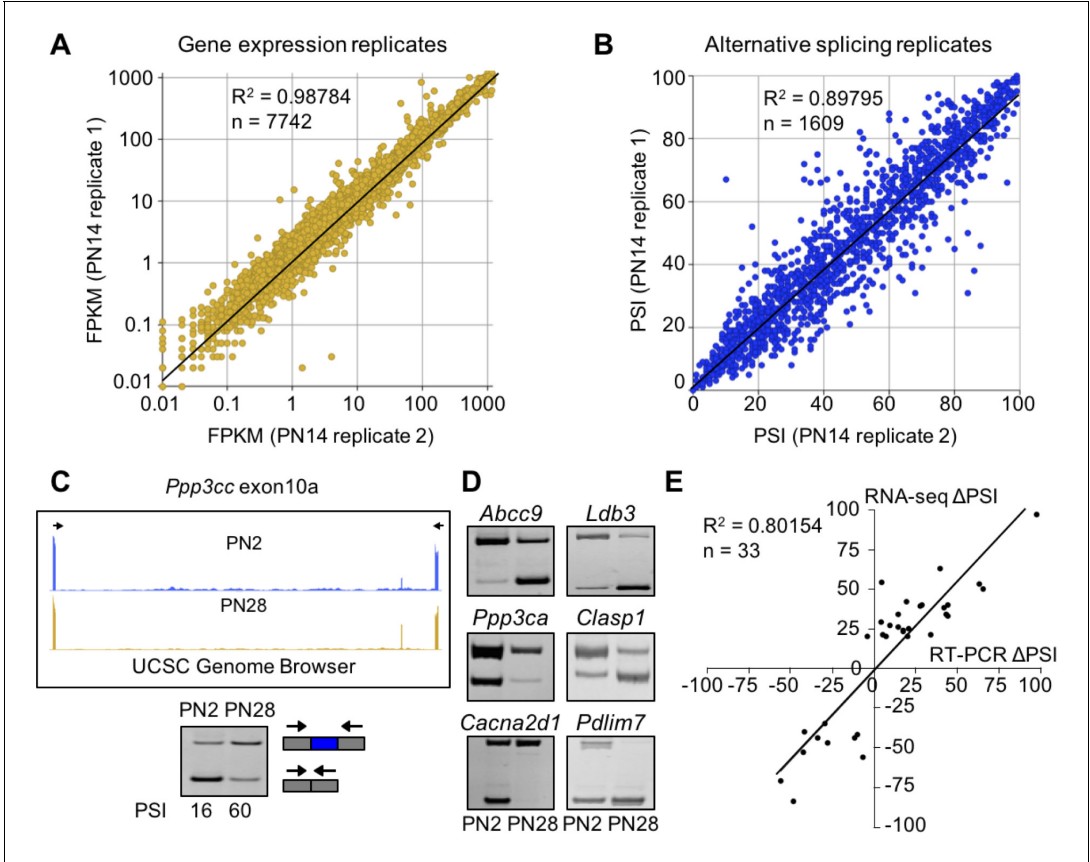

**Figure 1.** RNA-seq data is high quality and reproducible. (**A and B**) Biological replicates for PN14 were compared to analyze variation in gene expression (**A**) and alternative splicing (**B**) using Cufflinks and MISO, respectively. (**C**) RT-PCR to quantitate splicing use primers that anneal to the constitutive exons flanking an alternative exon, displayed on the UCSC Genome Browser (above) to determine the percent spliced in (PSI). (**D**) RT-PCR splice products for alternative splicing events comparing PN2 to PN28. (**E**) Plot comparing PN2 to PN28 RNA-seq ΔPSI values and ΔPSI values obtained by RT-PCR.

DOI: https://doi.org/10.7554/eLife.27192.002

The following figure supplement is available for figure 1:

**Figure supplement 1.** The vast majority of transcriptome changes observed in the RNA-seq data are from myofibers with a minimal contribution from satellite cells.

DOI: https://doi.org/10.7554/eLife.27192.003

changed in gene expression from E18.5 to PN2 and 1496 genes (25%) changed expression between PN28 and adult (*Figure 2A*).

The numbers of proliferating satellite cells decrease during early postnatal skeletal muscle development such that a portion of the transcriptome changes are likely to reflect changes in cell population rather than transitions within established myofibers. The RNA-seq data show that the expression of markers of activated satellite cells is relatively low even at PN2 while the changes in expression of myofiber markers (*Myog*, *Des*, and *Myh4*) are robust (*Figure 1—figure supplement 1*). These results are consistent with the contention that the dynamic transcriptome changes reflect transitions within established myofibers with minimal contributions from a changing satellite cell population.

To identify the timing of splicing transitions, we compared ΔPSIs of the four developmental intervals (*Figure 2B*, *Table 1*). Interestingly, 67% of the splicing events with a ΔPSI ≥ 15% undergo a transition during only one time interval (*Figure 2C*). Similar to differential gene expression, the interval between PN2 and PN14 has the largest number of splicing transitions. Of the 768 splicing events that occur between E18.5 and adult, 32% occur specifically between PN2 and PN14 while only 13% occur between E18.5 and PN2, 5% occur specifically between PN14 and PN28, and 16% occur

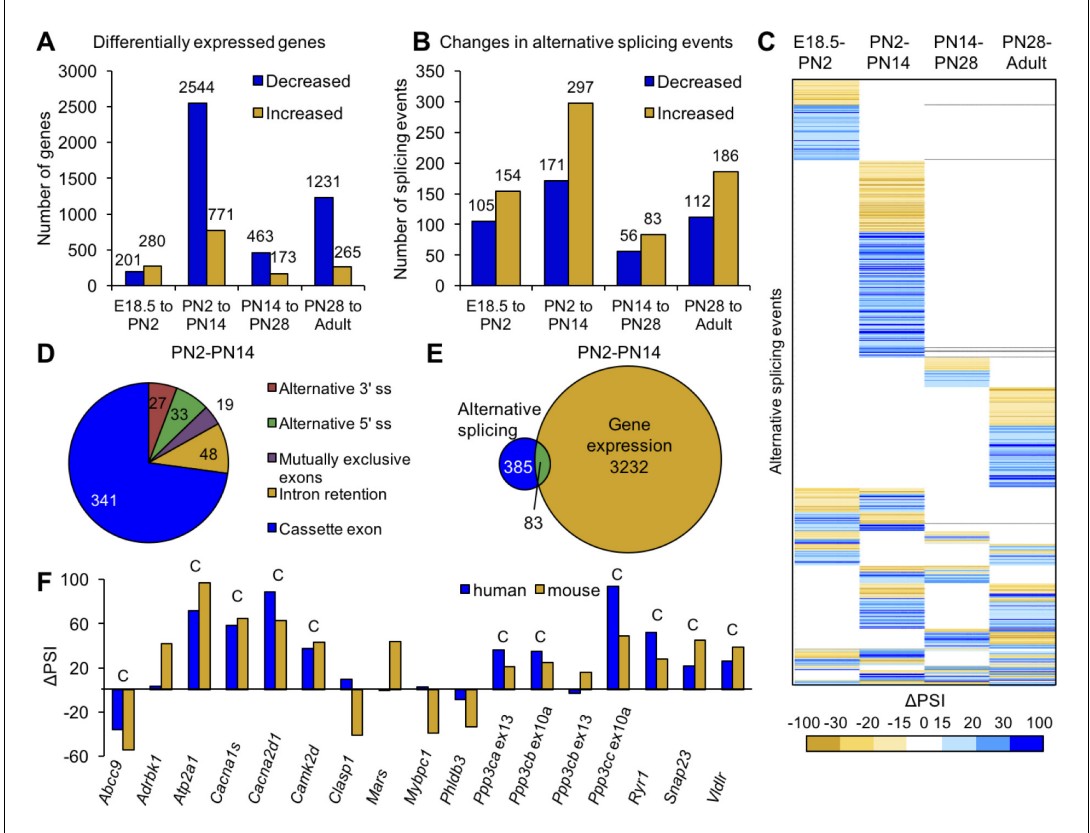

**Figure 2.** Postnatal gene expression and alternative splicing transitions in mouse skeletal muscle are largely independent, temporally restricted and conserved. (A) Genes with ≥2.0 fold increase or decrease in expression between E18.5-PN2, PN2-PN14, PN14-PN28 and PN28-Adult (*Table 1*). (B) Alternative splicing events with ΔPSI ≥± 15% between E18.5-PN2, PN2-PN14, PN14-PN28 and PN28-Adult. Decreased indicates more skipping of a splicing event during development, and increased indicates more inclusion of an alternative splicing event during development. (C) Heat map of alternative splicing transitions between four time intervals. Most gene expression and splicing transitions occur between PN2 and PN14. (D) Splicing patterns of events with 15% ΔPSI or greater between PN2 and PN14. (E) Venn diagram of genes with gene expression changes (2.0-fold or greater) compared to alternative splicing transitions (15% ΔPSI or greater) between PN2 and PN14. (F) Conservation of splicing transitions during mouse and human skeletal muscle development. The ΔPSI between PN2 to PN28 mouse gastrocnemius samples were compared to the ΔPSI between gestation week 22 to adult human skeletal muscle by RT-PCR. Events showing a 15% ΔPSI or greater in the same direction in mouse and human samples were scored as conserved (indicated by C).

DOI: https://doi.org/10.7554/eLife.27192.005

The following figure supplement is available for figure 2:

**Figure supplement 1.** Most changes in gene expression occur before PN14.
DOI: https://doi.org/10.7554/eLife.27192.006

specifically between PN28 and adult. These results indicate that there is enrichment for alternative splicing changes within the first 2 weeks after birth, 77% of which are cassette exons from PN2 to PN14 (*Figure 2D*).

While there are large numbers of transitions for both alternative splicing and gene expression between PN2 and PN14, there is little overlap in the genes that undergo these transitions. Of the genes that undergo alternative splicing transitions between PN2 and PN14, only 18% also showed differential gene expression indicating that the majority of alternative splicing changes are within genes that do not significantly change expression (*Figure 2E*). Of the 768 differential splicing events that change between E18.5 and adult, 31% do not exhibit differential gene expression between ED18.5 and adult and 58% of differential alternative splicing events are within genes that show differential expression in a different and single time interval (*Figure 2—figure supplement 1B*). This indicates that differential gene expression and alternative splicing are both dynamic throughout

postnatal development, but affect different sets of genes or, interestingly, affect the same genes at different developmental periods. Since the majority of splicing changes are within open-reading frames (ORFs) (see below), for many genes there is a major impact on protein isoform transitions rather than a change in gene output.

To determine the level of conservation of validated splicing transitions, we performed RT-PCR using RNA from PN2 and PN28 mouse skeletal muscle and human skeletal muscle RNA from 22 weeks gestation and adult. The alternative splicing events selected to test for conservation in humans were enriched for cassette exons found in calcium-handling genes since this functional category was strongly enriched (see below). Alternative splicing events were considered to be conserved if both the mouse and human had a ΔPSI of 15% or greater in the same direction. Of the 17 splicing events tested, 11 (65%) underwent a transition that was conserved (*Figure 2F*). Since all 11 events involved cassette exons that inserted or removed in-frame peptides, the results suggest that the different protein isoforms that result from the alternative splicing transitions have conserved physiological functions.

## In-frame alternative exons predominate in the ORF while out-of-frame alternative exons contain coding and untranslated regions

To determine the impact of alternative splicing on the protein isoforms expressed during postnatal development, we examined the distribution of cassette alternative exons within the spliced mRNAs and the effect of regulated splicing on the reading frame. For alternative exons with a ΔPSI of 15% or greater from PN2 to PN28 and in genes that contain at least three constitutive exons (223 exons), the relative position of the alternative exon along the length of mRNA was determined by dividing the exon number of the alternative exon by the total number of exons then multiplying by 100 to derive the exon order relative to the mRNA 5′ end (percent from the 5′ end). Alternative exons were also separated based on whether or not they are a multiple of three nucleotides since the latter change the reading frame. Alternative exons that are a multiple of three were predominantly found to maintain the ORF (129 exons of 135 considered) causing either an internal insertion or deletion of amino acids (*Figure 3A and B*). Six exons either created translation start or stop codons or altered the 5′ UTR. Alternative exons that are not a multiple of three (88 exons) were enriched near the 5′ or 3′ ends of the mRNA either exclusively within the UTRs (22 exons) or affecting the ORF to produce alternative N- or C-termini (66 exons) (*Figure 3C and D*). Therefore, the majority of alternative splicing transitions produce fetal and adult protein isoforms differing by internal peptide segments. Less common, but still prevalent, are transitions that produce different N- or C-termini. The least common are alternative exons that affect untranslated regions without affecting the reading frame.

## Genes that undergo postnatal splicing transitions during skeletal muscle development are enriched for calcium handling functions

Ingenuity analysis of genes that undergo changes in gene expression and alternative splicing between PN2 and PN28 identified essentially non-overlapping functional categories (*Figure 4*). Specifically, genes that undergo differential expression were enriched for associations with mitochondrial function while alternative splicing transitions were enriched for genes associated with calcium handling, endocytosis, and cell junction categories (*Figure 4A and B*). Although calcium-related categories included 90 genes for both gene expression and alternative splicing (75 for gene expression, 21 for alternative splicing), only six genes were found to have significant expression and splicing changes. *Table 2* lists the calcium handling genes that undergo ΔPSI ≥ 15 point change in alternative splicing between PN2 and PN28. Calcium handling is critical to striated muscle contractility and homeostasis and our results indicate that a large fraction of calcium handling genes undergo splicing transitions that affect the coding potential of these genes (*Table 2* and *Figure 5*). These results suggest an important role for alternative splicing transitions from fetal to adult isoforms in calcium handling genes during postnatal skeletal muscle development.

## CELF and MBNL proteins regulate distinct sets of calcium handling genes

CELF and MBNL proteins are involved in alternative splicing transitions during normal developmental and in skeletal muscle disease (*Giudice et al., 2014*; *Kalsotra et al., 2008*; *Lin et al., 2006*;

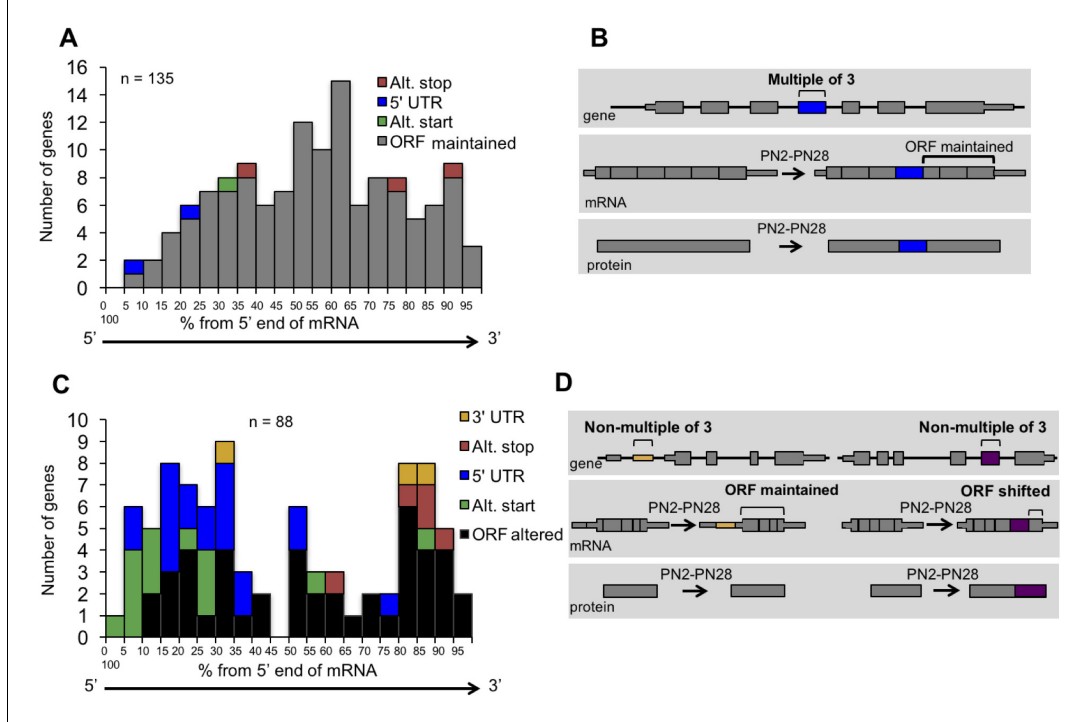

**Figure 3.** Distribution of cassette alternative exons within mRNAs. The graphs display the number of genes and the relative position of the alternative cassette exon from the 5' end of the spliced mRNA. The relative exon position is expressed as the percent of the total number of exons. The alternative exons analyzed (223 exons) have a ΔPSI of 15% or greater from PN2 to PN28 and contain at least three constitutive exons. (A) In-frame alternative exons that maintain the reading frame. Colors indicate whether the alternative exon contains a translational stop codon (red), start codon (green), only 3' UTR sequence (yellow), only 5' UTR sequence (blue), or if the exon is within the open reading frame (gray). (B) Representation of potential protein-coding consequences of in-frame alternative exons. (C) Out of frame alternative exons that shift the reading frame. (D) Representation of potential protein-coding consequences of alternative exons that are a non-multiple of three.

DOI: https://doi.org/10.7554/eLife.27192.007

*Konieczny et al., 2014*; *Dasgupta and Ladd, 2012*). Of the six *Celf* paralogs and three *Mbnl* paralogs in mice, *Celf1*, *Celf2*, *Mbnl1*, and *Mbnl2* are expressed in postnatal and adult skeletal muscle and therefore comprise the totality of CELF and MBNL activities during postnatal development. Western blot analysis of protein expression during postnatal development of gastrocnemius muscle demonstrated that Celf1, Celf2, and Mbnl2 protein levels decrease dramatically between PN7 and PN14 (*Figure 6A*). Mbnl1 protein expression is also reduced after PN7. Published results indicate that Mbnl1 undergoes translocation to the nucleus during mouse postnatal skeletal muscle development (*Lin et al., 2006*) that is likely to increase its effects on its splicing targets. The dramatic changes in expression of CELF and MBNL proteins during the first 2 weeks after birth show a strong correlation with a particularly dynamic period of splicing change and differential protein expression of these two families of splicing regulators.

To determine whether changes in Celf1 and Mbnl1 protein levels affect postnatally regulated alternative exons, we analyzed splicing of genes with postnatal splicing transitions in mouse skeletal muscle from our previously published skeletal muscle-specific tetracycline-inducible transgenic mice induced to overexpress human (h)CELF1 in adult animals (MDAFrtTA/TRECUGBP1 + dox vs. control MDAFrtTA +dox) and from adult *Mbnl1* knock out mice (*Mbnl1*$^{\Delta E3/\Delta E3}$ vs. control *Mbnl1*$^{+/+}$) (*Figure 6B–C* and *Figure 6—figure supplement 1*) (*Ward et al., 2010*; *Kanadia et al., 2003*). hCELF1 protein expression in induced bitransgenic gastrocnemius muscle was approximately 8-fold above endogenous levels as previously published (*Ward et al., 2010*) and Mbnl1 protein was not detected in *Mbnl1*$^{\Delta E3/\Delta E3}$ gastrocnemius used for RNA isolation (*Figure 6—figure supplement 1A*). Overexpression of hCELF1 and loss of Mbnl1 produced six and nine significant splicing changes, respectively (15 genes affected of the 18 tested). The 15 splicing events affected by either hCELF1

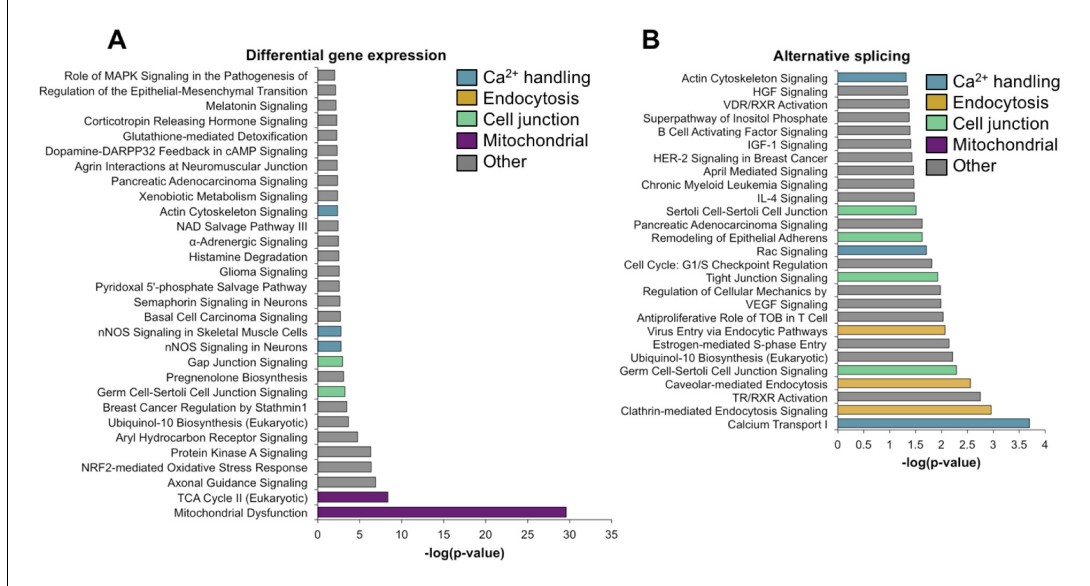

**Figure 4.** Gene ontology analysis for differential gene expression and alternative splicing. Ingenuity analysis was performed for both gene expression and alternative splicing, with a significance cut-off of –log(1.3). (A) Ingenuity analysis for gene expression differences between PN2 and PN28 (2-fold cut-off). The top 30 GO terms are displayed. For significant GO terms not shown, none of the highlighted, colored terms were present. (B) Ingenuity analysis for alternative splicing for genes with a 15% ΔPSI or greater between PN2 and PN28. All significant GO terms are shown.
DOI: https://doi.org/10.7554/eLife.27192.008

overexpression or loss of Mbnl1 reverted toward the splicing pattern observed in fetal muscle (*Figure 6—figure supplement 1B*) consistent with a response to a loss of CELF1 and a gain of MBNL1 activity during postnatal development. Only two genes, *Prkca* and *Ppp3cc* responded to both gain of hCELF1 and loss of endogenous Mbnl1. These results indicate that CELF and MBNL proteins are important contributors to regulated splicing within genes involved in calcium handling during postnatal skeletal muscle development.

## All three calcineurin A genes undergo fetal to adult protein isoform transitions during postnatal development

To determine the functional consequences of postnatal splicing transitions, we focused on the calcineurin A genes (*Ppp3ca*, *Ppp3cb*, and *Ppp3c*). Calcineurin is a calcium sensitive phosphatase affecting fiber type in skeletal muscle by dephosphorylating Nuclear Factor of Activated T-cells component (NFATC) proteins leading to NFATC nuclear translocation and NFATC-mediated transcriptional activation of a select subset of genes. It is a heterodimer containing a catalytic subunit, calcineurin A, and regulatory subunit, calcineurin B. All three calcineurin A paralogs expressed in skeletal muscle show postnatal splicing transitions with a ΔPSI of at least 15% (*Table 2*, *Figure 5*, and *Figure 7A*) while the two calcineurin B genes do not (data not shown). The functional consequences for the calcineurin A splicing events have not been characterized.

*Ppp3cb* and *Ppp3cc* each have a homologous 27 base pair alternative exon 10a with in-frame insertions encoding peptide segments with low-sequence identity (*Figure 7—figure supplement 1A*). *Ppp3ca* does not have a protein sequence equivalent to exon 10a. *Ppp3ca* and *Ppp3cb* each have an alternatively spliced 30 base pair exon 13 encoding nearly identical amino acid sequences (*Figure 7—figure supplement 1A*). Exon 13 for *Ppp3cc* is constitutively included and encodes a peptide sequence that is 40% and 50% identical to *Ppp3ca* and *Ppp3cb*, respectively (*Figure 7—figure supplement 1A*). All four alternative exons have increased inclusion during postnatal development. In addition, exons 10a of *Ppp3cb* and *Ppp3cc* are included specifically in adult skeletal muscle while exons 13 of *Ppp3ca* and *Ppp3cb* are included in other adult tissues including heart, testis, and brain (*Figure 7B*).

**Table 2.** Calcium handling genes with alternative splicing transitions during skeletal muscle development.
Listed are calcium-handling genes from the GO analysis (*Figure 3*). The ΔPSI from PN2 to PN28 are displayed along with the size of alternative exon, effect on the reading frame, the relative location of the exon, and predicted protein-coding consequence.

| Gene symbol | Gene | ΔPSI | Size of exon (bp) | In frame? | Alt. exon position | Effect on coding |
|---|---|---|---|---|---|---|
| Atp2a1 | ATPase, Ca ++ transporting, cardiac muscle, fast twitch | 97 | 42 | Y | 22 of 23 | C-term. |
| Atp2a3 | ATPase, Ca ++ transporting, ubiquitous | 20 | 73 | N | 21 of 22 | C-term. |
| Atp2b3 | ATPase, Ca ++ transporting, plasma membrane | −42 | 154 | N | 21 of 22 | C-term. |
| Cacna1s | Calcium channel, voltage-dependent, L type, alpha 1S subunit | 50 | 57 | Y | 29 of 44 | Insertion |
| Cacna2d1 | Calcium channel, voltage-dependent, alpha 2/delta subunit 1 | 53 | 57 | Y | 19 of 39 | Insertion |
| Calu | Calumenin | −27 | 194 | Y | three of 7 | Mutually exclusive |
| Camk2b | Calcium/calmodulin-dependent protein kinase II beta | 62 | 129 | Y | 13 of 17 | Insertion |
| Camk2d | Calcium/calmodulin-dependent protein kinase II delta | 34 | 89 | N | 19 of 20 | C-term. |
| Camsap1 | Calmodulin regulated spectrin-associated protein 1 | 25 | 33 | Y | five of 18 | Insertion |
| Cask | Calcium/calmodulin-dependent serine protein kinase | −44 | 69 | Y | 14 of 21 | Deletion |
| Chrne | Cholinergic receptor, nicotinic, epsilon (muscle) | −18 | 107 | N | 5 of 12 | Change of a.a. |
| Kcnn1 | Potassium intermediate/small conductance calcium-activated channel, subfamily N, member 1 | −35 | 111 | Y | 6 of 10 | Deletion |
| Mef2d | Myocyte enhancer factor 2D | −31 | 138 | Y | 4 of 12 | Mutually exclusive |
| Nfatc3 | Nuclear factor of activated T-cells, cytoplasmic, calcineruin-dependent 3 | 20 | 104 | N | 10 of 11 | C-term. |
| Ppp3ca | Protein phosphatase 3, catalytic subunit, alpha isozyme (calcineurin Aα) | 30 | 30 | Y | 13 of 14 | Insertion |
| Ppp3cb | Protein phosphatase 3, catalytic subunit, beta isozyme (calcineurin Aβ) | 25 | 27 | Y | 11 of 15 | Insertion |
| Ppp3cb | Protein phosphatase 3, catalytic subunit, beta isozyme (calcineurin Aβ) | 16 | 30 | Y | 14 of 15 | Insertion |
| Ppp3cc | Protein phosphatase 3, catalytic subunit, gamma isozyme (calcineurin Aγ) | 49 | 27 | Y | 11 of 15 | Insertion |
| Ryr1 | Ryanodine receptor 1 (skeletal) | 39 | 18 | Y | 83 of 106 | Insertion |
| Tnnt3 | Troponin T type 3 (skeletal, fast) | 22 | 41 | Y | 16 of 17 | Mutually exclusive |
| Trdn | Triadin | 18 | 60 | Y | 9 of 36 | Insertion |

DOI: https://doi.org/10.7554/eLife.27192.010

Calcineurin A contains a calcineurin B binding domain, calmodulin binding domain, phosphatase domain, and an auto-inhibitory domain (*Al-Shanti and Stewart, 2009*). Alternative exons 10a of *Ppp3cb* and *Ppp3cc* are adjacent to and upstream of the calmodulin-binding domain, and exons 13 of *Ppp3ca* and *Ppp3cb* are adjacent to and upstream of the auto-inhibitory domain (*Figure 7—figure supplement 1B*). These positions suggest a potential effect on the functions of these domains. In addition, the conservation of the postnatal transitions between mouse and human suggests that the isoform transition is functionally relevant to tissue remodeling during the fetal to adult transition (*Figure 2F*). Inclusion of calcineurin A alternative exons during postnatal development is not strongly affected by fiber type since the splicing transitions are similar between gastrocnemius and soleus muscles, which have different proportions of fast and slow fiber muscles (*Figure 7—figure supplement 1C*).

## Re-directed calcineurin A splicing reveals an isoform-specific effect on Nfatc3 activation

To determine the specific effects of calcineurin A redirected splicing on downstream signaling, we used the mouse C2C12 myogenic cell line. C2C12 cells transition from proliferative myoblasts to fused myotubes upon withdrawal of growth factors. Differentiating C2C12 cultures reproduce the developmental inclusion of exons 13 of *Ppp3cb* and *Ppp3ca* but not *Ppp3cc* and only weak inclusion of exons 10a of *Ppp3cb* and *Ppp3cc* (*Figure 8—figure supplement 1A*). We used morpholino

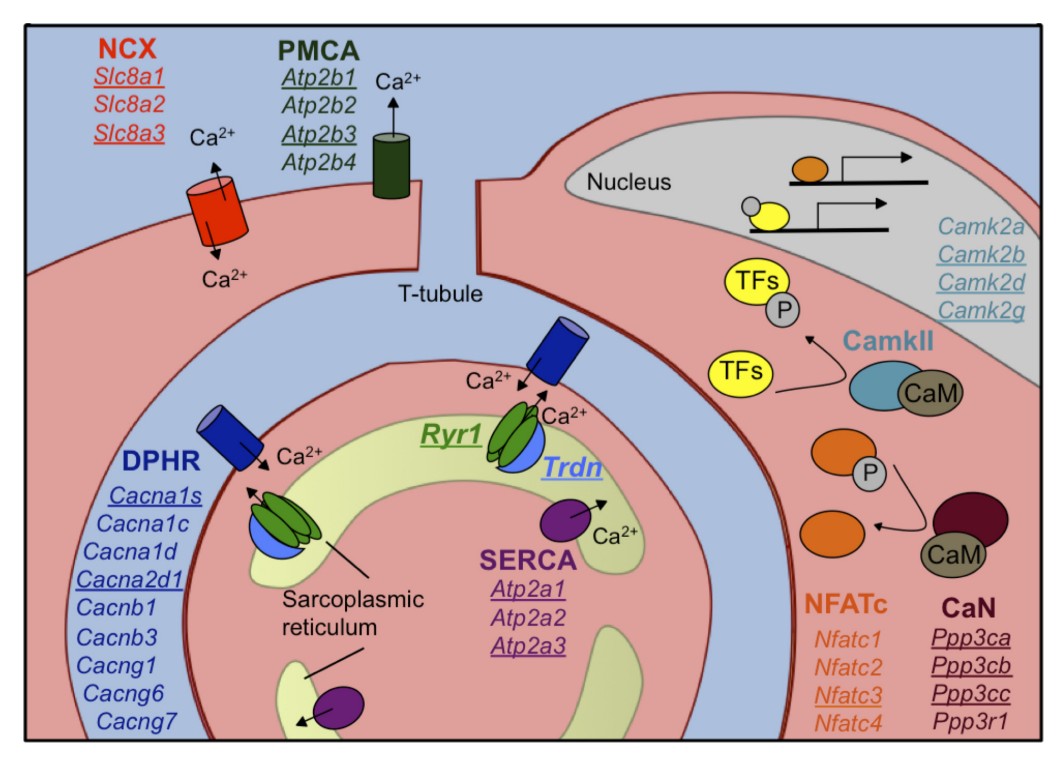

**Figure 5.** Calcium handling genes that undergo alternative splicing transitions in postnatal skeletal muscle development. Diagram of calcium handling genes that are expressed in skeletal muscle. Genes with 15% or greater ΔPSI from PN2 to PN28 are underlined and include members of several calcium channels: NCX (sodium calcium exchanger), PMCA (plasma membrane $Ca^{2+}$-ATPase), SERCA (sarco/endoplasmic reticulum $Ca^{2+}$-ATPase), and RYR (ryanodine receptor); triadin (Trdn) which associates with Ryr1, junctin (Asph) and FKBP12 (Fkbp1a). Signaling cascades that are affected by alternative splicing include $Ca^{2+}$/camodulin (CaM)-dependent calcineurin (CaN) and calmodulin-dependent protein kinase II (CamkII) along with the downstream transcription factor NFATC. NFATC and transcription factors (TFs) regulated by CamkII activate genes for hypertrophy and fiber type specification.
DOI: https://doi.org/10.7554/eLife.27192.009

antisense oligonucleotides (ASO) to redirect splicing of exons 13 of *Ppp3cb* and *Ppp3ca* through C2C12 differentiation (*Figure 8A*). Immunofluorescence staining was then used to determine the effects on NFATC family members that are expressed in differentiated myotubes. Nfatc3 protein showed significantly increased nuclear localization in myotubes with calcineurin A redirected splicing while Nfatc1 localization was unchanged (*Figure 8B*, *Figure 8—figure supplement 1B*). *Nfatc2* mRNA levels are very low in differentiated C2C12 based on our RNA-seq analysis (*Singh et al., 2014*) and immunofluorescence staining showed no clear change in signal from myotube nuclei (*Figure 8B*, *Figure 8—figure supplement 1B*). The Nfatc3 immunofluorescence signal was validated by knockdown in C2C12 myotubes (*Figure 8—figure supplement 1C*). These results demonstrate that expression of the endogenous fetal isoforms of *Ppp3ca* and *Ppp3cc* is sufficient to activate translocation of one of the three Nfatc proteins, Nfatc3.

To determine the physiological impact of the calcineurin A splicing transitions in adult skeletal muscle, we used ASOs delivered into the FDB foot pad muscle to redirect all four calcineurin A exons to the fetal splicing pattern of exon skipping. This approach allows testing the functions of specific endogenous protein isoforms without changing the overall expression level. We found that the efficiency of redirected splicing remains high for at least 4 weeks and 3 weeks provides sufficient time for the muscle to recover from the ASO delivery procedure (*Giudice et al., 2016a*). Three weeks after delivery of redirecting ASOs or non-targeting control ASOs, redirected splicing was assayed by RT-PCR and all four exons were found to have undergone a nearly complete switch to the fetal pattern while non-targeting ASOs had no effect (*Figure 9A and B*). Four non-targeted developmental splicing events were assayed (*Atp2a1*, *Cacna1s*, *Prkca*, *Ppp1r12b*) and none showed

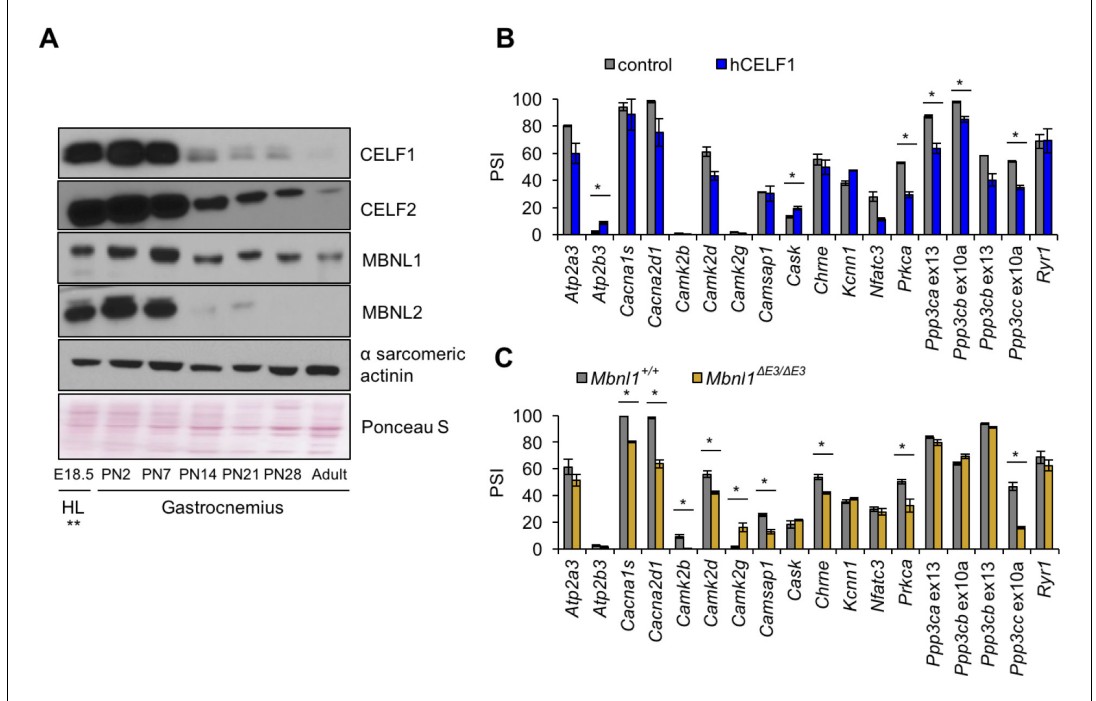

**Figure 6.** Postnatal down-regulation of CELF and MBNL alternative splicing regulators promote splicing transitions of calcium handling genes. (**A**) Western blot of Celf1, Celf2, Mbnl1, and Mbnl2 during gastrocnemius postnatal development. **All E18.5 samples are from hindlimb (HL) except Celf1 blot which is E18.5 gastrocnemius. Ponceau S and α sarcomeric actinin serve as loading markers. (**B**) Comparing PSI of control mice (MDAFrtTA + dox) and human CELF1 overexpressing mice (MDAFrtTA/TRECUGBP1 + dox) (C57BL6/DBA;FVB). (**C**) Comparing PSI of wild type and Mbnl1 KO mice, Mbnl1$^{\Delta E3/\Delta E3}$ (FVB). Single asterisk (*) denotes p<0.05 using student T-test, n = 3 mice per group. Displayed are mean with SD bars.

DOI: https://doi.org/10.7554/eLife.27192.011

The following figure supplement is available for figure 6:

**Figure supplement 1.** CELF1 overexpression and Mbnl1 KO effect on calcium-handling splicing.

DOI: https://doi.org/10.7554/eLife.27192.012

significant differences following ASO injection, demonstrating the absence of non-specific effects (**Figure 9B**).

To assess physiological changes from redirected calcineurin A splicing, we measured parameters of force generation and calcium handling in FDB muscle bundles three weeks after ASO delivery. Redirected splicing resulted in a significant prolongation of twitch half relaxation time and a strong trend toward increased peak twitch force and peak calcium, although these changes did not reach statistical significance (**Figure 9C**). There was no difference in twitch time to peak force or calcium for a twitch (**Figure 9—figure supplement 1**). Following peak tetanic stimulation (150 Hz), we also found a significant prolongation of the half relaxation time and a trend toward increased tetanic peak force and peak calcium (**Figure 9D**) (see Discussion).

To determine whether expression of the fetal calcineurin A isoforms in adult FDB muscle affected Nfatc transcriptional activity, we assayed expression of known Nfatc-targeted genes. *Myh1* showed a significant change while *Myh4*, and *Myh8* genes showed a trend that did not yh1, *Myh4*, and *Myh8* genes. Knock down of Nfatc3 in C2C12 resulted in decreased *Myh8* mRNA levels and a trend toward reduced *Myh1* mRNA expression (**Figure 8C**). Since these genes are up-regulated in response to redirected calcineurin splicing in FDB muscle and down-regulated upon loss of NFATC3 activity in C2C12 cultures, these results strongly suggest that the fetal isoforms of at least *Ppp3ca* and *Ppp3cb* have intrinsically higher phosphatase activity than the adult isoforms, resulting in Nfat3 nuclear localization and subsequent Nfatc3 transcriptional changes iB\xBDse activity than the adult isoforms, resulting in Nfat3 nuclear localization and subsequent Nfatc3 transcriptional changes in skeletal muscle producing a prolonged relaxation time. These results also identify Nfatc3 as the family member that is likely to be primarily responsive to the fetal isoforms.

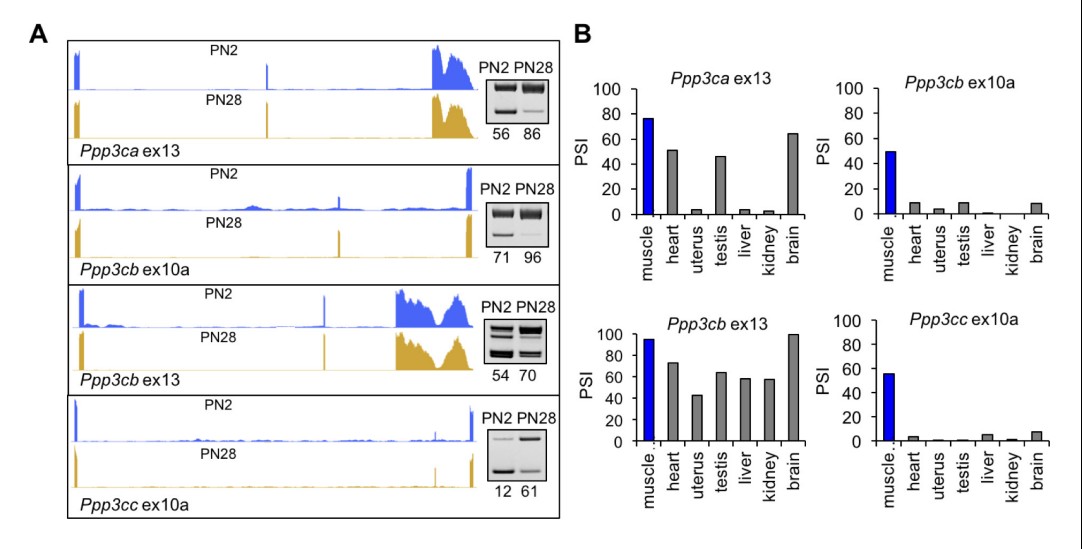

**Figure 7.** Calcineurin A splicing during PN development in different tissues. (**A**) UCSC Genome Browser displays of calcineurin A alternative exons (*Ppp3ca* ex13, *Ppp3cb* ex10a, *Ppp3cb* ex13, *and Ppp3cc* ex10a) side-by-side with RT-PCR of RNA from PN2 and PN28. (**B**) RT-PCR comparing inclusion of calcineurin A alternative exons in diverse adult tissues (BioChain tissue array mRNA).

DOI: https://doi.org/10.7554/eLife.27192.013

The following figure supplement is available for figure 7:

**Figure supplement 1.** Alignment of calcineurin A alternative exons.

DOI: https://doi.org/10.7554/eLife.27192.014

## Discussion

Our analysis of postnatal skeletal muscle development indicates that both transcriptional and post-transcriptional changes, particularly alternative splicing, are highly dynamic and particularly during the first 2 weeks after birth. Seventy-nine percent of genes that undergo postnatal splicing transitions do not show significant changes in gene expression within the same time interval indicating a separation of regulated splicing and the regulation of mRNA steady state levels. That different gene sets are regulated at the level of splicing and mRNA levels has been found during other periods of physiological change including T cell activation, muscle differentiation and heart development (*Giudice et al., 2014*; *Ip et al., 2007*; *Singh et al., 2014*; *Trapnell et al., 2010*). These results support a mechanism of regulation that involves transitions of fetal to adult protein isoforms rather than a change in total gene output for a relatively large subset of genes. Genes regulated by differential expression or alternative splicing were enriched for different functional categories: mitochondrial function for differentially expressed genes and calcium handling, endocytosis, and cell junctions for genes that undergo alternative splicing transitions. The results for alternative splicing during postnatal skeletal muscle development are similar to postnatal heart development in which vesicular trafficking genes were enriched among those regulated by alternative splicing (*Giudice et al., 2014*; *Kalsotra et al., 2008*).

Calcium regulates signal transduction, muscle contraction, and cellular homeostasis. Our results indicate that calcium channels, calcium-dependent phosphatases, and calcium-dependent kinases are alternatively spliced soon after birth. Several calcium handling genes have been established as alternatively spliced such as Serca1 (*Atp2a1*) and the ryanodine receptor 1 (*Ryr1*) while most, such as calcineurin A, have not been characterized (*Kimura et al., 2007*; *Kimura et al., 2009*; *Periasamy and Kalyanasundaram, 2007*). We show that all three calcineurin A genes (*Ppp3ca*, *Ppp3cb* and *Ppp3cc*) undergo fetal to adult protein isoform transitions within the first 2 weeks after birth. Redirected splicing to the fetal patterns demonstrate that the fetal protein isoforms do not fulfill the functional requirements of adult skeletal muscle tissue in that both a significant increase in time to relaxation and a physiologically sizable, although not statistically significant, increase in peak force and calcium. Calcium activates muscle contraction and the peak isometric force is dictated by

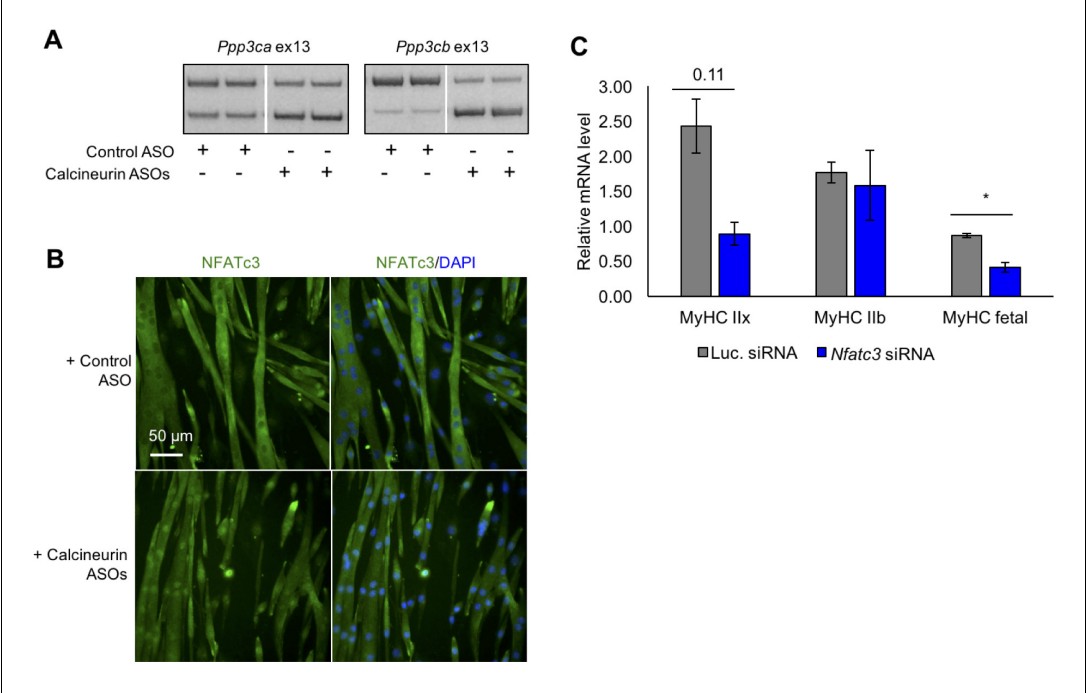

**Figure 8.** Redirected calcineurin A splicing in differentiated C2C12 myotubes. (**A**) RT-PCR of redirected splicing of calcineurin A events in differentiated C2C12 cells. Two of the four calcineurin A splicing events are present in C2C12. For calcineurin A ASO, 9 µM *Ppp3ca* ex13 3'ss and 15 µM *Ppp3cb* ex13 5'ss morphlinos were delivered, and for control ASO, 24 µM standard control morpholino were delivered. (**B**) Immunofluorescence of Nfatc3 in differentiated C2C12 after morpholino delivery. (**C**) mRNA levels of Nfatc targets after knockdown of Nfatc3 in differentiated C2C12 cells. Relative mRNA levels are standardized to Mrps7 mRNA levels; single asterisk (*) denotes p<0.05 significance by student T-test, n = 3 biological replicates. Displayed are mean with SD bars.

DOI: https://doi.org/10.7554/eLife.27192.015

The following figure supplements are available for figure 8:

**Figure supplement 1.** Analysis of calcineurin A splicing during C2C12 differentiaion, Nfatc1 and Nfatc2 protein localization in C2C12, and validation of Nfatc3 antibody.

DOI: https://doi.org/10.7554/eLife.27192.016

**Figure supplement 2.** Sequences for antisense morpholinos and primers.

DOI: https://doi.org/10.7554/eLife.27192.017

the peak amplitude of the calcium transient. Therefore, the increased peak calcium upon redirected splicing would lead to the observed increase in peak isometric force. We propose that the mechanisms of altered adult muscle function are due at least in part to altered regulation of Nfatc proteins by calcineurin A. We show that redirected splicing of *Ppp3ca* and *Ppp3cb* pre-mRNA in differentiated C2C12 myotubes leads to nuclear localization of Nfatc3, two of the three Nfatc proteins expressed in these cultures. Our results are consistent with previous results showing that calcineurin overexpression in C2C12 causes nuclear localization of Nfatc3 (*Delling et al., 2000*). Calcineurin-dependent nuclear localization of Nfatc activates slow-fiber-specific genes (*Delling et al., 2000*; *Chin et al., 1998*; *Liu et al., 2001*). We found that redirected splicing of the three calcineurin genes upregulated *Myh1* in FDB muscle and knock down of Nfatc3 in differentiated C2C12 cultures resulted in down-regulation of *Myh1*. Two sequences similar to the predicted NFATC consensus site are found in the promoter of MyHC IIx (*Chin et al., 1998*). and our results are consistent with up regulation of *Myh1* by increased Nfatc activity. Overall, however, redirected calcineurin splicing had a modest effect on Nfat transcription targets in vivo. A possible explanation is that NFATC proteins do not act alone but rather regulate transcription in combination with multiple factors. We forced expression of the endogenous calcineurin fetal isoforms in an otherwise adult transcriptional environment and it is likely that co-regulators and other factors required for full NFATC activity are either not expressed or are not in the appropriate active state. Overall, our results support a model in

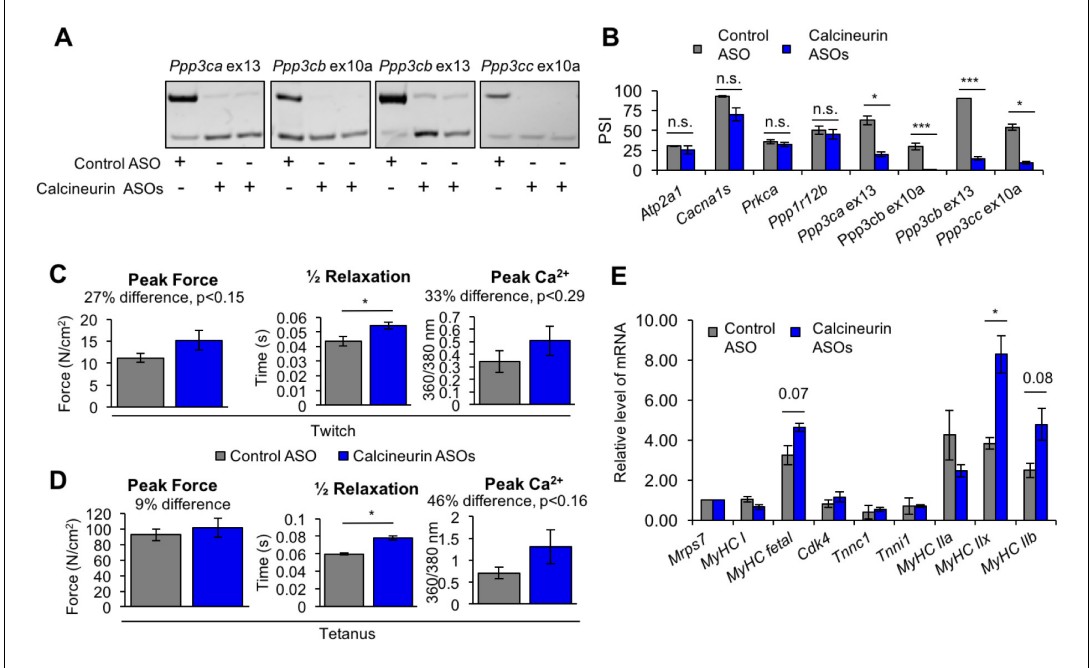

**Figure 9.** Redirected calcineurin A splicing in adult FDB muscle. (**A**) Confirmation of redirected splicing in adult FDB muscle by RT-PCR. ASO morpholinos were administered to the FDB muscle (*Figure 8—figure supplement 2*). (**B**) RT-PCR of targeted (calcineurin A) and control (*Atp2a1*, *Cacna1s*, *Prkca*, and *Ppp1r12b*) splicing events. Single asterisk (*) denotes p<0.05, *** denotes p<0.001, and n.s. denotes no statistical difference by student T-test, n = 4 mice per group. Displayed are mean with SD bars. (**C**) Force and calcium analysis after twitch stimulus. Peak force and half relaxation time were measured for force and peak calcium for calcium. Single asterisk (*) denotes p<0.05 significance by student T-test, n = 3–4 mice per group. Displayed are mean with SD bars. (**D**) Force and calcium analysis after tetanus stimuli. Peak force and half relaxation time were measured for force and peak calcium for calcium. Single asterisk (*) denotes p<0.05 significance by student T-test, n = 3–4 mice per group. Displayed are mean with SD bars. (**E**) Relative mRNA levels of mRNAs from Nfatc target genes in FDB muscle by RT-PCR. Single asterisk (*) denotes p<0.05 and marked are transcriptional targets nearing significance by student T-test, n = 3 mice per group. Displayed are mean with SD bars.

DOI: https://doi.org/10.7554/eLife.27192.018

The following figure supplement is available for figure 9:

**Figure supplement 1.** Not significant (by student T-test) time to peak of force and calcium for control ASO and calcineurin A ASOs, n = 3–4 mice per group.

DOI: https://doi.org/10.7554/eLife.27192.019

which the fetal calcineurin isoforms have higher intrinsic phosphatase activity on NFATC3 thereby promoting its activation.

Protein segments encoded by conserved tissue specific alternative exons are enriched for disordered domains more likely to affect protein-protein interactions and contain post-translation modification (PTM) sites (*Ellis et al., 2012*; *Buljan et al., 2012*). Affecting protein-protein interactions or PTMs could be a means by which calcineurin splicing events affect calcineurin activity or stability. The alternative exons could affect the ability of calcineurin A to be regulated by calmodulin or through auto-inhibition. A splice variant, calcineurin Aβ1, has a C-terminal truncation so that the auto-inhibitory domain is missing. Overexpression of this splice variant improved cardiac function in mice (*Felkin et al., 2011*; *Gómez-Salinero et al., 2016*). Calcineurin Aβ1 was not abundant in our RNA-seq analysis of skeletal muscle development, but it does give insight to functional consequences of an alternative isoform.

We also investigated the mechanisms that regulate a subset of the dramatic splicing transitions during postnatal development. We identified abrupt changes in CELF and MBNL protein levels during postnatal skeletal muscle development that temporally correlate with the splicing transitions that occur by PN14. We used hCELF1 overexpressing and *Mbnl1* knockout mice to show that 72% of the calcium handling genes tested that exhibit alternative splicing transitions respond to these proteins and are likely regulated by the natural transitions in CELF and MBNL activity that occur during

postnatal development. Investigations using *Celf1*[-/-] mice identified a network of CELF1-dependent splicing that correlate with heart defects in early postnatal development (*Giudice et al., 2016b*). CELF1 is also likely to be important for muscle regeneration as CELF1 protein increases after muscle injury and Celf1 splicing targets revert to fetal splicing patterns (*Orengo et al., 2011*). MBNL proteins negatively regulate embryonic stem cell-like patterns of alternative splicing, and overexpression of MBNL1 in embryonic stem cells produces a differentiation-like cell alternative splicing pattern (*Han et al., 2013*). Although MBNL1 and MBNL2 protein levels decrease postnatally, the activity of at least MBNL1 is proposed to increase due to nuclear localization during postnatal development (*Lin et al., 2006*). While we have identified splicing events that respond to changes in CELF1 and MBNL1 protein abundance, these events are not necessarily direct targets for these two RNA-binding proteins. In addition the responsiveness of these splicing events to Celf1 and Mbnl1 does not rule out regulation by other RNA-binding proteins.

A growing number of global RNA-seq analyses are revealing the extent to which conserved alternative splicing transitions are critical for tissue function (*Giudice et al., 2014*; *Singh et al., 2014*; *Buljan et al., 2012*; *Kroeze et al., 2017*; *Vernia et al., 2016*). In this work, we identify the dynamic transcriptome changes during skeletal muscle postnatal development and demonstrate the functional significance for the calcineurin A splicing transitions. The results demonstrate the importance of understanding the functional differences between fetal and adult protein isoforms and the contribution of these proteins to tissue remodeling to adult function.

## Materials and methods

### Animals
Skeletal muscle tissues were isolated from FVB wild type, MDAFrtTA/TRECUGBP1, and Mbnl1[ΔE3/ΔE3] mice. We followed NIH guidelines for use and care of laboratory animals approved by Baylor College of Medicine Institutional Animal Care and Use Committee.

### Skeletal muscle isolation and RNA extraction
Animals were anesthetized and euthanized either by decapitation (neonatal) or cervical dislocation (older than PN10), and gastrocnemius muscles were removed. Sex determination for animals PN7 and younger was confirmed by PCR using primers to Actin and Sry genes (sequences in *Figure 8— figure supplement 2*). Tissue samples were flash frozen with liquid nitrogen. Total RNA was prepared using the RNeasy fibrous tissue mini kit (Qiagen).

### RNA-seq
Illumina TruSeq protocols were used to prepare libraries using total RNA (2 ug) from gastrocnemius of E18.5, PN2, PN14, PN28, and 22-week (adult) animals. The cDNA was created using the fragmented 3'-poly(A)-selected portion of total RNA and random primers. To generate the libraries, the blunt ended fragments of cDNA were attached to adenosine to the 3'-end and ligated with unique adapters to the ends. The ligated products were amplified by PCR for 15 cycles. Libraries were quantified and fragment size assessed by the NanoDrop spectrophotometer and Agilent Bioanalyzer, respectively. The libraries were amplified by qPCR to determine the concentration of adapter-ligated fragments using a Bio-Rad iCycler iQ Real-Time PCR Detection System and a KAPA Library Quant Kit. The library (11 pM) was loaded onto a flow cell and amplified by bridge amplification using Illumina cBot equipment. On a HiSeq Sequencing system, a paired-end 100-cycle run was used to sequence the flow cell.

### Computational processing and bioinformatics of RNA-seq data
For the RNA-seq alignment, paired-end RNA-seq reads were aligned to the mouse genome (mm9) using TopHat 2.0.5 (*Trapnell et al., 2009*). For the differential gene expression analysis, RSEM was used to count the number of fragments mapped into RefSeq gene models, and edgeR was used to call differentially expressed genes with a false discovery rate less than 0.05 (RSEM: accurate transcript quantification from RNA-Seq data with or without a reference genome) (*Trapnell et al., 2010*; *Robinson et al., 2010*). Gene expression was quantified by FPKM. For differential alternative splicing analysis, isoform levels (PSI) and Bayes factors (BI) were measured by MISO (*Katz et al., 2010*) with

ΔPSI >= 0.1 and BI >= 10. For both differential gene expression and alternative splicing analysis, data were analyzed through the use of QIAGEN's Ingenuity Pathway Analysis (IPA, QIAGEN Redwood City, www.qiagen.com/ingenuity).

## Alternative splicing validations and human conservation by RT-PCR

Human skeletal muscle RNA samples were obtained for fetal 22 week (BioChain [R1244171-50]) and adult (Clontech [636534]). Various adult mouse tissue total RNA samples (muscle, heart, uterus, testis, liver, kidney, and brain) were obtained from BioChain. For mouse and human RNAs, 2.5 ug of RNA was used for reverse transcription (RT). RT was performed by High Capacity cDNA RT Kit (Applied Biosystem) followed by PCR (GoTaq DNA Polymerase, Promega). RT-PCR products were separated by 6% PAGE. PCR reactions involved: 95°C for 3 min, 25–30 cycles of 95°C for 45 s, 55°C for 45 s, 72°C for 45 s, and 72°C for 5 min. RNA-seq data was used identify alternatively splicing regions and primers (Sigma) that anneal to the constitutive flanking exons were designed. Primers are listed in *Figure 8—figure supplement 2*. Ethidium bromide-stained RT-PCR bands were quantified by Kodak Gel logic 2000 and Carestream Software. PSI values were calculated by densitometry using the equation: PSI = 100 X [Inclusion band/(Inclusion band +Skipping band)]. RT-PCRs were repeated by at least two technical replicates.

## Western blotting

FVB wild-type gastrocnemius tissues were lysed in HEPES-sucrose buffer (10 mM HEPES pH 7.4, 0.32 M sucrose, 1 mM EDTA and protease inhibitors) using Bullet blender (Next Advance) and SDS (final concentration of 1%) was added before sonication (3 min at 75 V for 30 s on and 30 s off). The samples were centrifuged for 10 min at 12,000 rpm at 4°C. Supernatants were transferred to new tubes, and samples were diluted in loading buffer (100 mM Tris-HCl pH 6.8, 4% SDS, 0.2% bromophenol blue, 20% glycerol, 200 mM β-mercaptoethanol) then boiled for 3 min. Pierce Compat-able BCA protein assay kit (Thermo Scientific) was used to quantify protein concentration after the addition of loading buffer. For each sample 40 μg of protein was loaded into a 10% SDS-PAGE gel. Proteins were transferred to membranes and blocked with 5% milk/0.1% Tween-PBS buffer for 1 hr, washed, and incubated overnight at 4°C with 5% milk/Tween-PBS buffer diluted primary antibodies: mouse monoclonal anti-CELF1 clone 3B1 (1:1000), CELF2 (Santa Cruz Biotechnology [sc-47731]— 1:1000), Mbnl1 (LifeSpan Biosciences [LS-C30810]–1:1000), Mbnl2 (Santa Cruz Biotechnology [sc-136167]–1:1000), rabbit polyclonal anti-sarcomeric α-actinin (Abcam #ab72592-1:2,000). CELF1 and CELF2 monoclonal antibodies were conjugated to HRP using Abnova Peroxidase Labeling Kit – NH2. Non-conjugated primary antibodies were incubated the following day for 1 hr at room temperature with secondary antibodies: goat anti-mouse IgG light chain-specific HRP conjugated (Jackson Immunoresearch [#115-035-174]–1:10000) and goat anti-rabbit IgG HRP conjugated (Invitrogen, [#621234]–1:5000). Super Signal West Pico Chemilumiescent Substrate kit (Thermo Scientific) was used for developing.

## ASO injection in vivo

Animal protocols were approved by IACUC at Baylor College of Medicine. FVB wild-type adult mice were anesthetized by isoflurane in a chamber and moved to a nose cone for injections. First, the FDB muscle was pretreated with hyaluronidase (0.5 mg/ml, 10 μl) injected subcutaneously. After 2 hr, morpholino ASOs (80 μg *Ppp3ca* ex13 3'ss, 20 μg *Ppp3cb* ex10a 3'ss, 20 μg *Ppp3cb* ex13 5'ss, and 80 μg *Ppp3cc* ex 10a 3'ss or 200 μg of standard control, 15 μl) (Gene-Tools, sequence in *Figure 8—figure supplement 2*) were injected followed by electroporation. Electroporation parameters were 150 V, 20 s duration, no delay, 1 Hz, train 0.5, and duration 400. Mice were assayed for splicing redirection by RT-PCR and other downstream assays 3 weeks after injection.

## In vitro calcium and force assays

FDB lateral and medial muscle bundles were dissected away leaving the central muscle bundle and tendon. The central muscle bundle tendon was attached to a fixed hook and the other to a force transducer. The muscle was placed in physiological saline solution, continuously gassed with 95% $O_2$/5% $CO_2$ at 25°C, and loaded with 5 μM Fura 4 F AM (Invitrogen). After 30 min, samples were rinsed with fresh solution and then allowed to de-esterify for 30 min. The optimal muscle length ($L_o$)

and voltage ($V_{max}$) were adjusted to induce maximum twitch force. Twitch and tetanic force were measured at 1 and 150 Hz with pulse and train durations of 0.5 and 250 ms, respectively. Fura 4 F AM excitation (360/380 nm) and emission (510 nm) were monitored simultaneously with force-frequency characteristics. After stimulation, muscle length was measured and fiber bundles were trimmed of excess muscle and connective tissue, blotted dry, and weighed. Muscle weight and $L_o$ were used to estimate cross sectional area and to calculate absolute forces expressed as N/cm$^2$ (*Close, 1972*). To determine intracellular calcium changes during FDB stimulation, the 360/380 nm ratio was calculated.

## Cell culture

C2C12 cells were maintained in DMEM with 10% FBS in six-well tissue culture plates. To differentiate cells, cultures were grown to 100% confluency, and media was changed to DMEM with 2% horse serum. To redirect splicing in C2C12, 9–15 µM morpholinos were delivered by Endo-Porter (Gene-Tools) at 50% confluency. Morpholinos and Endo-Porter were added to differentiation media after cells reached 100% confluency. Differentiated C2C12 myotubes were collected at day 4. For one biological replicate, three separate wells of control ASO and calcineurin ASOs (total of six wells) were collected for downstream experiments. Cell culture experiments were performed in at least three biological replicates.

## Immunofluorescence

Undifferentiated or differentiated C2C12 cells were grown in six-well tissue culture plates containing glass coverslips. Cells were fixed with 4% paraformaldehyde in PBS for 15 min at room temperature. Fixed cells were washed with PBS 2x and permeabilized using 0.2% triton-X in PBS for 10 min. Cells were then blocked in 5% BSA in PBS at room temperature for 1 hr and incubated overnight in primary antibody in 5% BSA in PBS at 4°C. The cells were washed 3x with PBS followed by Alexa Fluor-conjugated secondary antibody incubation for 1 hr, washed 3x with PBS, DAPI stained for 5 min, and washed 3x with PBS. Deconvolution microscopy was performed by GE Healthcare Inverted Deconvolution/Image Restoration Microscope.

## Statistics

For statistical analysis, at least three samples were pooled together to determine average and variance. Error bars represent the standard deviation, and student T-test was used determine significant with $p > 0.05$. With sample size indicated in figure legends, it was confirmed experimental sample size gave at least power of 0.80 with an $\alpha$ of 0.05. Biological replicates refer to individual mice or separate wells for cell culture. Technical replicates refer to RT-PCR replicates from the same cDNA or RNA.

## Acknowledgements

We thank Jimena Giudice (UNC) for her advice on RNA-seq analysis and FDB electroporation, Joshua Sharpe (BCM) for help on comparing alternative splicing between human and mice by RT-PCR, Kathleen Manning (BCM) for help with RNA-seq analysis, and Charles Thornton for *Mbnl1$^{\Delta E3/\Delta E3}$ mice*. This project was supported by the Genomic and RNA Profiling Core at BCM and the expert assistance of the core director, Dr. Lisa D White, Ph.D. The Integrated Microscopy Core at BCM also supported this project with funding from NIH (NCI-CA125123, NIDDK-56338-13/15), Cancer Prevention Research Institute of Texas (RP150578), and John S Dunn Gulf Coast Consortium for Chemical Genomics.

## Additional information

### Funding

| Funder | Grant reference number | Author |
| --- | --- | --- |
| National Institutes of Health | R01AR045653 | Thomas A Cooper |

| | | |
|---|---|---|
| Muscular Dystrophy Association | RG4205 | Thomas A Cooper |
| National Institutes of Health | R01HL045565 | Thomas A Cooper |
| National Institutes of Health | R01AR060733 | Thomas A Cooper |
| National Institutes of Health | T32 HL007676 | James Anthony Loehr |
| National Institutes of Health | R01HG007538 | Wei Li |
| National Institutes of Health | R01CA193466 | Wei Li |
| National Institutes of Health | R01AR061370 | George Gerald Rodney |

The funders had no role in study design, data collection and interpretation, or the decision to submit the work for publication.

## Author contributions

Amy E Brinegar, Conceptualization, Data curation, Formal analysis, Validation, Investigation, Visualization, Methodology, Writing—original draft, Project administration, Writing—review and editing; Zheng Xia, Data curation, Software, Formal analysis, Writing—review and editing; James Anthony Loehr, Data curation, Formal analysis, Methodology, Writing—review and editing; Wei Li, Supervision, Writing—review and editing; George Gerald Rodney, Formal analysis, Supervision, Writing—review and editing; Thomas A Cooper, Conceptualization, Formal analysis, Supervision, Funding acquisition, Writing—original draft, Project administration

## Author ORCIDs

Thomas A Cooper http://orcid.org/0000-0002-9238-0578

## Ethics

Animal experimentation: All animals were handled following the NIH Guidelines for Use and Care of Laboratory Animals that were approved by the Institutional Animal Care and Use Committee (IACUC) at Baylor College of Medicine, protocol AN-1682).

## Decision letter and Author response

Decision letter https://doi.org/10.7554/eLife.27192.024
Author response https://doi.org/10.7554/eLife.27192.025

# Additional files

## Supplementary files

• Transparent reporting form
DOI: https://doi.org/10.7554/eLife.27192.020

## Major datasets

The following dataset was generated:

| Author(s) | Year | Dataset title | Dataset URL | Database, license, and accessibility information |
|---|---|---|---|---|
| Amy E Brinegar, Zheng Xia, Wei Li, Thomas A Cooper | 2017 | Differentially modulated transcriptomes of mouse skeletal muscle (gastrocnemius) during postnatal development. | https://www.ncbi.nlm.nih.gov/geo/query/acc.cgi?acc=GSE108402 | Publicly available at NCBIGene Expression Omnibus (accession no: GSE108402) |

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
