## [Decision Letter]

Thank you for submitting your article "Extensive alternative splicing transitions during postnatal skeletal muscle development are required for Ca^2+^ handling" for consideration by *eLife*. Your article has been reviewed by three peer reviewers, and the evaluation has been overseen by a Reviewing Editor and James Manley as the Senior Editor. The reviewers have opted to remain anonymous.

The reviewers have discussed the reviews with one another and the Reviewing Editor has drafted this decision to help you prepare a revised submission.

In this study, the authors investigate transcriptomic changes that accompany post-natal muscle development. Using RNA-seq to profile five developmental stages, the authors identify large numbers of gene expression and alternative splicing changes that accompany muscle development. In both instances, the most pronounced changes occur during the first two weeks after birth. Similar to what has been demonstrated for other developmental processes, the authors show that the events that change at the level of expression and splicing are largely non-overlapping and affect distinct GO categories. Interestingly, alternative splicing appears to target multiple genes involved in calcium handling. To assess the implications for muscle function, the authors focus their attention on four alternative exons located in three calcineurin A genes (*Ppp3ca, Ppp3cb*, and *Ppp3cc*). The authors show that reverting calcineurin A splicing to fetal-like patterns using antisense oligos leads to changes in Nfatc cellular localization in C2C12 cells. They conclude that the corresponding transcriptional changes are responsible for the altered FDB muscle relaxation times they observe in mice.

Understanding the functional consequences of developmental splicing changes is of considerable interest. Given the central role of calcium in muscle function, the concentration of splicing events in diverse calcium-handling genes identified in this study may thus critically affect muscle function through a variety of mechanisms. The authors provide initial evidence suggesting that alternative splicing in calcineurin impacts Nfatc localization and thus, transcription of Nfatc-target genes. The authors are asked to further support these conclusions, as well as to address additional concerns listed below, prior to publication.

Essential revisions:

1) The authors conclusions are based on bringing together results from two separate experiments performed in two independent systems: they observe changes in Nfatc localization in C2C12 cells following splicing reversion of two calcineurin exons and they observe changes in Nfatc-dependent gene expression in mice following splicing reversion of all four calcineurin exons. Although the effects on Nfatc3 localization in Figure 8 appear dramatic, the quantification indicates that the increase in the nuclear fraction of Nfatc2 and Nfatc3 following ASO treatment is only ~2 and 3%, respectively. It is unclear if these changes are sufficient to impact Nfatc-dependent gene expression. As such, it is recommended that the authors test expression of Nfatc targets specifically in this system to determine whether these changes are sufficient. Also, while the 10a exons do not change during C2C12 differentiation, it is possible that forcing their skipping in this system in addition to exons 13 would increase the magnitude of the observed effect. The complementary experiment would be to assay Nfatc localization in mice treated with ASOs.

2) While the authors conclude that alternative splicing leads to changes in Nfatc-dependent transcription in mice, they report significant transcriptional changes in only one Nfatc-target gene (MyHC IIx) of 9 genes tested. The authors do not provide an explanation for why they think only one Nfatc gene is affected. It is recommended that they determine whether the observed change is Nfatc-dependent by repeating the experiment in the context of Nfatc knockdown (perhaps more feasible in C2C12 cells).

3) The role of CELF and MBNL proteins in the P2-P14 alternative splicing changes should be clarified. The correlation between expression of CELF1, CELF2, MBNL1 and MBNL2 and the identified splicing changes could be explained better. Figure 6 indicates that the levels of these RBPs decrease sharply after P7 and remain low into the adulthood. In Figure 2, what are the ages of mice used for P2-P14 samples? Are these samples from several mice? If the data are divided further into P2-P7 vs. P8-P14, are bigger splicing differences observed? In Figure 2, the data could be better explained, at least in legends. What are the events shown in the bottom part, changes observed in multiple stages? Also, what is the fraction of the alternative splicing events shown in Figure 2 for P2-P14 (top half of the diagram) that have binding sites of CELF and MBNL in their pre-mRNAs? What about Figure 6? The question is whether these RBPs are directly involved in regulating the splicing events. There are also two concerns regarding data shown in Figure 6. First, what is the rationale of decreasing MBNL1, as its level also decreases after P7? Second, the protein levels of the over-expression and gene deletion experiments should be shown in this study. The authors cited their own paper for the over-expression study and a paper on gene deletion from the Swanson laboratory. It is important to know the protein levels in the experiments in the current study. Is calcineurin a target of CELF or MBNL? Are there binding sites for these proteins?

---

## [Author Response]

Essential revisions:

*1) The authors conclusions are based on bringing together results from two separate experiments performed in two independent systems: they observe changes in Nfatc localization in C2C12 cells following splicing reversion of two calcineurin exons and they observe changes in Nfatc-dependent gene expression in mice following splicing reversion of all four calcineurin exons. Although the effects on Nfatc3 localization in Figure 8 appear dramatic, the quantification indicates that the increase in the nuclear fraction of Nfatc2 and Nfatc3 following ASO treatment is only ~2 and 3%, respectively.*

We agree that the quantitation in Figure 8 seems to under represent the strong change in nuclear accumulation of Nfatc3 that is quite apparent visually and indicated a change in Nfatc2 nuclear localization that is not visually apparent. Given the discrepancy between the clear shift of Nfatc3 compared to Nfatc2, we re-evaluated the details of the experiment and conclude that we do not have sufficient confidence in the microscopy quantitation to present it for publication. Nfatc2 and Nfatc3 localization was quantified comparing nuclear to cytoplasmic signals in five to six randomly selected images. The analysis was of all nuclei in the field including multinucleated myotubes (the cells of interest) and residual undifferentiated mononucleated cells. This raises one issue of background from non-differentiated cells. The Nfatc3 antibody was validated by siRNA knock down (Figure 8—figure supplement 1) so we are confident of the Nfatc3 reagent. Our RNA-seq data [*Mol. Cell* 55, 592–603 (2014)] shows that Nfatc2 mRNA levels are very low in our C2C12 cultures and our RNA-seq data also shows that Nfatc1 is 2.7 times lower than Nfatc3. Therefore, it is very likely that Nfatc3 is by far the major Nfatc isoform expressed in our differentiated C2C12 cultures and its nuclear localization is strongly affected by redirected splicing of the calcineurin splicing events. We have therefore focused on the clear shift to nuclear accumulation of Nfatc3 presented in Figure 8.

It is unclear if these changes are sufficient to impact Nfatc-dependent gene expression. As such, it is recommended that the authors test expression of Nfatc targets specifically in this system to determine whether these changes are sufficient.

The same suggestion to assay Nfatc targets after a Nfatc knock down was also raised below in point #2 and we agree that this is an important test. We performed knocked down of Nfatc3 in differentiated C2C12 cultures followed by analysis of Nfatc transcriptional targets to “determine whether the observed change is Nfatc-dependent” as suggested in point #2. We focused on MyHC IIx, MyHC IIb, and MyHC fetal as Nfatc targets in C2C12 since the mRNA levels of these genes were significantly affected or trended toward significant change in response to ASO redirected splicing in mouse FDB muscle (Figure 9). Nfatc3 knock down showed a significant down regulation of MyHC fetal mRNA levels and trending down regulation of MyHC IIx (p = 0.11). These new results are now presented in Figure 8.

To summarize, we saw down regulation of MyHC IIx (trending) and MyHC fetal (significant) in Nfatc3 knock down in C2C12 (Figure 8) and we saw up regulation of MyHC IIx (significant) and MyHC fetal (trending) when calcineurin splicing is redirected in FDB muscle (Figure 9). Therefore, our in vivo and in vitro results are wholly consistent with a model in which redirected calcineurin splicing results in activation and nuclear localization of Nfatc (Figure 8) correlating with increased expression of Nfatc targets.

Also, while the 10a exons do not change during C2C12 differentiation, it is possible that forcing their skipping in this system in addition to exons 13 would increase the magnitude of the observed effect.

Of the three calcineurin genes, *Ppp3ca* does not have an alternative exon 10a and the level of inclusion of exons 10a in *Ppp3cb* and *Ppp3cc* remain low in differentiated cultures compared to exon 13 (RT-PCR analysis for splicing of these exons is now presented in Figure 8—figure supplement 1 as suggested by reviewers, below). While there could be a small effect of forcing exon skipping of exon 10a in *Ppp3cb* and *Ppp3cc* in C2C12, we felt that this would not substantially add to the dramatic effect of Nfatc3 nuclear localization shown in Figure 8.

The complementary experiment would be to assay Nfatc localization in mice treated with ASOs.

While this result would add another connection between in vitro and in vivo experimental systems, we were unable to perform the experiment due to technical and timing issues. However, our new results show that knock down of Nfatc3 in differentiated C2C12 cultures affects the same targets that are affected by redirected splicing in mouse FDB muscle. The specificity of the result is worth highlighting: redirected splicing of only 4 exons in 3 endogenous calcineurin genes in muscle tissue correlates with increased expression of two Nfatc target genes in the tissue. These same genes are also down regulated by knock down of Nfatc3 in differentiated C2C12 cultures. Furthermore, redirected splicing of calcineurin genes in C2C12 produces nuclear localization of Nfatc3 consistent with the observed up-regulation of the Nfatc targets in FDB muscle in which calcineurin splicing is redirected. We believe that these results strongly support the role for Nfatc activation in response to calcineurin splicing both in the vivo and in vitro experimental systems.

*2) While the authors conclude that alternative splicing leads to changes in Nfatc-dependent transcription in mice, they report significant transcriptional changes in only one Nfatc-target gene (MyHC IIx) of 9 genes tested. The authors do not provide an explanation for why they think only one Nfatc gene is affected.*

One explanation for the low number of Nfatc transcription targets affected is the fact that Nfatc transcription factors function in combination with multiple factors. We forced expression of the endogenous calcineurin fetal isoforms in an otherwise adult transcriptional environment and it is likely that co-regulators and other factors required for full Nfatc activity are either not expressed or are not in the appropriate active state. We have briefly mentioned this point in the Discussion.

It is recommended that they determine whether the observed change is Nfatc-dependent by repeating the experiment in the context of Nfatc knockdown (perhaps more feasible in C2C12 cells).

As discussed above, we have performed this experiment and the results have strengthened our conclusions. We thank the reviewer for the suggestion.

*3) The role of CELF and MBNL proteins in the P2-P14 alternative splicing changes should be clarified. The correlation between expression of CELF1, CELF2, MBNL1 and MBNL2 and the identified splicing changes could be explained better.*

Thank you for pointing out the lack of clarity. We reworded the section and also found that we had cited the wrong supplementary figure that likely added to the confusion.

Figure 6 indicates that the levels of these RBPs decrease sharply after P7 and remain low into the adulthood. In Figure 2, what are the ages of mice used for P2-P14 samples? Are these samples from several mice? If the data are divided further into P2-P7 vs. P8-P14, are bigger splicing differences observed?

In Figure 2, the samples designated as P2-P14, for example, means that RNA-seq data from one time point (PN2) was compared to RNA-seq data from a second time point (PN14) so each RNA-seq sample is from one time point and not a pool of different time points. The details for the numbers and genders of mice used for each time point are presented in Table 1 and the text and Figure legend now makes this designation clear and clarifies that the RNA-seq data in Figure 2 is from samples shown in Table 1. Analysis was not further divided to compare PN2 to PN7 and PN7 to PN14 since there was not a PN7 time point used for RNA-seq.

In Figure 2, the data could be better explained, at least in legends. What are the events shown in the bottom part, changes observed in multiple stages?

In panels A, B and C of Figure 2, the designation E18.5 to PN2 or PN2 to PN14, as examples, refers to comparisons of RNA-seq data collected at two different time points (e.g., 18.5, PN2, PN14, PN28 and adult). This designation does not reflect pooled samples of different stages. We have clarified this in the figure legend and in the description of the data in the text.

Also, what is the fraction of the alternative splicing events shown in Figure 2 for P2-P14 (top half of the diagram) that have binding sites of CELF and MBNL in their pre-mRNAs? What about Figure 6? The question is whether these RBPs are directly involved in regulating the splicing events.

We analyzed CLIP data for CELF1 and MBNL1 from mouse skeletal muscle and differentiated C2C12 [*Cell* 150, 710–724 (2012), *Genome Res.* 25, 858–871 (2015)]. We analyzed CLIP data for CELF1 from adult mouse skeletal muscle and for MBNL1 from differentiated C2C12 cultures since the MBNL1 data from C2C12 was much better than from muscle. We searched for CELF1 and MBNL1 binding sites within 100, 500, 1000, 1500, and 2000nt of the regulated exons in Figure 2 cassette exons that change at least ∆PSI 20 percentage points between PN2 and PN14). We found that of the 203 cassette exons, 39 and 52 exons were enriched for local CELF1 or MBNL1 binding sites, respectively within 1000 nt of the exon.

With regards to 18 genes listed in Figure 6, two of the six genes significantly responsive to CELF1 were associated with CELF1 CLIP tags and three of the nine genes responsive to loss of Mbnl1 were associated with Mbnl1 CLIP tags. There were also several genes that contained CELF1 and/or MBNL1 CLIP tags that were not responsive to either protein.

While it is gratifying that there is enrichment for CLIP tags for both CELF1 and MBNL1 among postnatally regulated cassette exons, we feel this does not substantially add to the manuscript. The data is consistent with the proposal that, analyzed as a group, a subset of exons that transition during postnatal development are direct targets but it is difficult to go beyond this general statement without direct functional validation of individual genes. In particular, with regards to the 18 genes listed in Figure 6, the general lack of correlation between binding sites and responsiveness is fraught with possible false positives and false negatives and might not be truly reflective of direct versus indirect effects. We have included this CLIP data for the reviewers in our response as excel file (CLIP_CELF1_MBNL1.xlsx) showing the number and positions of CLIP tags. Which of the splicing events that respond to changes in CELF or MBNL activities are direct targets is certainly of interest, but the main point for us is that there are substantial changes in the activities of the CELF and MBNL families during postnatal muscle development and there are resultant splicing consequences that have the potential to impact tissue physiology.

There are also two concerns regarding data shown in Figure 6. First, what is the rationale of decreasing MBNL1, as its level also decreases after P7?

Immunofluorescence results in mouse skeletal muscle from Charles Thornton’s lab showed that Mbnl1 translocates from the cytoplasm to the nucleus after birth [Lin, X. *et al.,* 2006]. Therefore, while we show reduced Mbnl1 protein expression in adult compared to early postnatal muscle, Mbnl1 nuclear activity is likely to increase postnatally. In addition, Mbnl1 protein is low but readily detectable in adult and furthermore loss of Mbnl1 in the Mbnl1 knock out line used for our molecular analysis has phenotypes produced by splicing abnormalities including myotonia due to failure to express the adult splicing pattern of the muscle specific chloride gene. Therefore, the level of Mbnl1 in adult muscle is functionally relevant.

Second, the protein levels of the over-expression and gene deletion experiments should be shown in this study. The authors cited their own paper for the over-expression study and a paper on gene deletion from the Swanson laboratory. It is important to know the protein levels in the experiments in the current study.

As requested by the reviewer, we have included western blots showing the level of CELF1 protein overexpression in induced transgenic compared to control muscle in Figure 6—figure supplement 1. The protein extracts for the western blots are from the same tissues from which RNA was extracted for the RT-PCR results shown in Figure 6. This figure also includes western blots for Mbnl1 that did not detect protein in the homozygous knock out animals. As previously demonstrated by our lab [*Hum. Mol. Genet.* 19, 3614–3622 (2010)], CELF1 was overexpressed approximately 8-fold in animals induced using the same conditions as was used for our studies and is shown in Figure 6—figure supplement 1.

Is calcineurin a target of CELF or MBNL? Are there binding sites for these proteins?

All three calcineurin genes responded to CELF1 overexpression and only *Ppp3cc* responded to loss of Mbnl1 (Figure 6). The CLIP analysis noted above found CELF1 CLIP tags near the exons 13 of *Ppp3cb* and *Ppp3ca*. While *Ppp3cb* exon 13 showed a trend in response to CELF1 overexpression the change did not reach statistical significance. *Ppp3ca* had one MBNL1 CLIP tag within 500 nucleotides of exon 13 that did not respond to Mbnl1 knock out. Therefore, there is not a strong correlation between binding detected by CLIP analysis in adult muscle and responsiveness of calcineurin splicing. However, as noted above, CLIP analysis is informative for a survey of a large number of genes but requires additional analyses when investigating the question of direct regulation for individual genes. We conclude that the results are too preliminary to rule out direct regulation.